# Disruptions to and Innovations in HPV Vaccination Strategies within Safety-Net Healthcare Settings Resulting from the COVID-19 Pandemic

**DOI:** 10.3390/healthcare11172380

**Published:** 2023-08-24

**Authors:** Samantha Garcia, Michelle Shin, Kylie Sloan, Emily Dang, Carlos Orellana Garcia, Lourdes Baezconde-Garbanati, Lawrence A. Palinkas, Benjamin F. Crabtree, Jennifer Tsui

**Affiliations:** 1Department of Population and Public Health Sciences, Keck School of Medicine, University of Southern California, Los Angeles, CA 90032, USA; sg_196@usc.edu (S.G.); ksloan@usc.edu (K.S.); emilydan@usc.edu (E.D.); corellan@usc.edu (C.O.G.); baezcond@usc.edu (L.B.-G.); 2Department of Child, Family, and Population Health Nursing, School of Nursing, University of Washington, Seattle, WA 98195, USA; mbyshin@uw.edu; 3Norris Comprehensive Cancer Center, University of Southern California, Los Angeles, CA 90033, USA; 4USC Suzanne Dworak-Peck School of Social Work, University of Southern California, Los Angeles, CA 90089, USA; palinkas@usc.edu; 5Robert Wood Johnson Medical School, Rutgers University, New Brunswick, NJ 08901, USA; crabtrbf@rwjms.rutgers.edu; 6Rutgers Cancer Institute of New Jersey, New Brunswick, NJ 08901, USA

**Keywords:** qualitative research, depth interviews, medically underserved, implementation strategies, safety-net settings

## Abstract

The COVID-19 pandemic disrupted healthcare delivery within safety-net settings. Barriers to and facilitators of human papillomavirus (HPV) vaccination during the pandemic can inform future HPV vaccine strategies for underserved communities. Qualitative interviews (*n* = 52) between December 2020 and January 2022 in Los Angeles and New Jersey were conducted with providers, clinic leaders, clinic staff, advocates, payers, and policy-level representatives involved in the HPV vaccine process. Using the updated Consolidated Framework for Implementation Research we identified (1) outer setting barriers (i.e., vaccine hesitancy driven by social media, political views during the pandemic) and facilitators (e.g., partnerships); (2) inner setting clinic facilitators (i.e., motivation-driven clinic metrics, patient outreach, vaccine outreach events); (3) individual characteristics such as patient barriers (i.e., less likely to utilize clinic services during the pandemic and therefore, additional outreach to address missed vaccine doses are needed); (4) innovations in HPV vaccination strategies (i.e., clinic workflow changes to minimize exposure to COVID-19, leveraging new community partnerships (e.g., with local schools)); and (5) implementation strategies (i.e., multisectoral commitment to HPV goals). Pandemic setbacks forced safety-net settings to develop new vaccine approaches and partnerships that may translate to new implementation strategies for HPV vaccination within local contexts and communities.

## 1. Introduction

The COVID-19 pandemic disrupted routine healthcare delivery, especially in historically marginalized communities. Structural barriers included loss of or change in health insurance coverage due to increased unemployment and reduced appointment availability as a result of workforce shortages and closures [1], affecting adolescent human papillomavirus (HPV) vaccination. The American Cancer Society’s 2021 HPV VACs Impact Report estimated that the United States (US) experienced between three and four million missed doses of the HPV vaccine due to the pandemic [2]. Prior to the pandemic, HPV vaccine hesitancy was attributed to complex multilevel factors, including individual (e.g., vaccine beliefs on safety and effectiveness), interpersonal (e.g., provider recommendation), and community factors (e.g., clinic environment, accessibility of appointment) [3,4]. Between March and August 2020, US HPV vaccine initiation rates were only 23% of the rates in 2018 and 2019 during the same time frame [5]. To prevent a similar disruption during a future pandemic and to inform vaccination efforts, there is an urgent need to understand how the pandemic impacted safety-net healthcare settings and vaccine-related partners working with these contexts and to identify the barriers and facilitators of HPV vaccine delivery and uptake. Individual barriers from the COVID-19 pandemic led to increased HPV vaccine delay and refusal [6,7], disrupted the delivery of routine well-child visits [8,9], and led to widespread misinformation and politicalization of vaccines, contributing to overall vaccine hesitancy [10]. Our prior study found that approximately 19% of parents from racial and ethnic minority communities report being highly hesitant towards the HPV vaccine [11]. Thus, in addition to multilevel factors known to impact HPV vaccination, consequences of the pandemic further complicate current approaches to HPV vaccine implementation. To prevent a similar disruption during a future pandemic and to inform vaccination efforts, there is an urgent need to understand how the pandemic impacted medically underserved communities and clinic settings to identify the barriers and facilitators of HPV vaccine delivery and uptake.

Evidence-based strategies (EBS) focused on clinic settings (e.g., patient reminders, provider communication trainings, provider reminders) have shown to improve HPV vaccination rates [12,13,14,15,16]. However, the pandemic likely disrupted such efforts due to stay-at-home orders, increased staff turnover, and a reduced priority for HPV vaccination given COVID-19 prevention and treatment efforts, reversing recent advancements in HPV vaccination in the United States. As priorities shift towards catch-up vaccination, community and clinic perspectives on pandemic disruptions to uptake and strategies to overcome them are needed, particularly within healthcare safety-net settings serving high-risk communities. In this study, we examined the perspectives of clinic members and community members, during the pandemic, to understand adaptations to HPV vaccine EBSs and new emerging strategies to deliver HPV vaccines among adolescents in medically underserved communities and safety-net healthcare settings. For the purposes of this study, we follow the Institute of Medicine’s definition of “health care safety-net” as a mission-driven system that provides substantial healthcare and services to uninsured, Medicaid, and underserved populations [17]. Our research question was: how did the COVID-19 pandemic impede and/or facilitate vaccination strategies that can support HPV vaccination among adolescents?

## 2. Materials and Methods

### 2.1. Participants and Procedures

This qualitative study is part of a larger mixed methods study funded by the National Cancer Institute (R37CA242541), where we recruited clinic- and community-level participants to investigate experiences with EBSs for HPV vaccination within safety-net primary care settings. Recruitment and data collection from this qualitative study has been reported elsewhere [18]. Briefly, we conducted 52 in-depth virtual interviews participants in greater Los Angeles (LA), California, and the state of New Jersey (NJ) over a 14-month period (December 2020 to January 2022). Participants were purposively sampled participants using snowball techniques to represent diverse clinic members (i.e., providers, clinic leaders, clinic staff) and community members (i.e., advocates, policy-level representatives, payers) that have insight into the HPV vaccination process among adolescents in safety-net healthcare settings. All participants were 18 years of age and older. Purposive sampling ensured that we were able to reach diverse study participants that worked in large and small clinics. We used an iterative purposive sampling approach until saturation was achieved [19]. In comparison to random sampling, which supports generalizability to larger populations, purposive sampling is used in qualitative studies to select participants with rich information and insight into a phenomenon of interest [19].

### 2.2. Data Collection 

Interviews were conducted virtually via Zoom in English. The Practice Change Model (PCM) [20] informed semi-structured interview guides, and one-on-one interviews lasted approximately 30 min to one hour. Interview guides sought to understand the (i) motivation of participants to implement (or change) strategies to increase HPV vaccination; (ii) clinic resources to support HPV vaccination strategies; (iii) outside motivation for implementing HPV vaccination strategies; and (iv) opportunities to improve on HPV vaccination strategies. An example interview question was, “How is COVID-19 impacting HPV vaccination in your region and the work you do with clinics and providers?”. Recordings from interviews were transcribed verbatim by a third-party transcription service. All transcripts were de-identified prior to analysis. 

### 2.3. Ethical Considerations

The Institutional Review Board at the University of Southern California and Rutgers University determined this study was exempt. We received verbal consent from all participants before conducting all interviews. Participants received 50 USD gift cards for completing the interview. Interview recordings were stored on a password-protected computer. Recordings were transcribed verbatim and de-identified.

### 2.4. Analysis 

Atlas.ti version 9 was used to analyze 52 transcripts. Using three phases [21], coders first individually immersed themselves in the data and iteratively identified three emergent codes related to the impact of COVID-19 on HPV vaccination through a series of meetings. Meetings consisted of discussing open coding and memo-taking. Second, two coders conducted focused coding [20] to organize COVID-19 themes into inner (clinic-based) and outer settings (community-based). Third, two coders then organized COVID-19 themes according to the updated Consolidated Framework for Implementation Research (CFIR) [22] domains including outer setting, inner setting, individuals, innovation, and implementation processes that emerged. Additional coding details to ensure high validity and reliability have been reported elsewhere [18]. 

## 3. Results

Nine themes emerged related to the impact of the COVID-19 pandemic on barriers and facilitators to HPV vaccine delivery (Table 1). Themes were organized using the updated CFIR domains [22] to address our research question: how did the COVID-19 pandemic impede and/or facilitate vaccination strategies that can support HPV vaccination among adolescents?

### 3.1. Outer Setting Domain 

Within the CFIR outer setting domain, policy and advocacy participants discussed (1) how the pandemic lowered HPV vaccine uptake and limited their ability to meet with providers and (2) the impact of the politicization of vaccines and vaccine misinformation increasing the barriers to HPV vaccination. 

**Theme 1: Policy and advocacy groups collaborating with the medical community throughout the COVID-19 pandemic to improve HPV vaccination rates.** One advocate shared the importance of spreading awareness of COVID-19’s impact on cancer prevention services including the drop in HPV vaccine rates to clinicians practicing in marginalized communities: 

“We had specifically reached out to the clinician community—pediatricians, family practice physicians, internists—with pretty descriptive messaging about what the current state [low HPV vaccine rates during the pandemic], and recommendations for fixing it.” (LA, Policy Representative)

For this policy representative, informing physicians and clinical staff was an important first step in addressing declining rates during the pandemic. 

Clinic staff and provider participants acknowledged a drop in HPV vaccine rates during the pandemic and spoke about the importance of continued efforts to catch up on adolescent vaccinations. Outer setting participants also discussed the importance of modifying training to continue educating providers on the best communication messages with parents about the HPV vaccine. One policy representative mentioned the pandemic led to the adaptation of HPV vaccine communication trainings from in-person presentations to webinars: 

“So, the [HPV vaccine] webinar [for providers], “Give it Your Best Shot”, it became an actual webinar where we didn’t even have to go in and provide a presentation anymore…Because of COVID now, my presentations were turned into webinars or live webinars.” (NJ, Advocate)

**Theme 2: The COVID-19 vaccine has been politicized and perpetuated misinformation on social media, which may also impact HPV vaccination rates.** Another emergent theme included the influence social media has on vaccine decisions. Inner setting participants described encounters with parents and adolescents addressing COVID-19 vaccine concerns that stemmed from misinformation, specifically on social media. A clinic staff person stated:

“I feel like when people come in with information, a lot of it is through social media. Whether it’s correct information or misinformation…When people are saying, ‘Well, I don’t want, you know, the COVID vaccine because I saw this on Facebook’—or whatever. And then usually that’s—you know, so that’s a social media response.” (LA, Provider)

Inner setting participants described social media as a platform for political dialogue and an echo chamber related to vaccines. Exposure to vaccine misinformation may vary across users and political party preferences on vaccination could expose some users to a higher degree of anti-vaccine material. Participants in this study suspected the politicization of the COVID-19 vaccine may increase HPV vaccine delay and refusal. 

“Over the last 10–15 years, you know, it’s gone through phases. And now we’re at a low phase where people have trust in vaccines, and a lot of it has become political. And I think that the whole thing that’s going on with the COVID vaccine has really made vaccines a political issue…So, they’re falling into the same trap for all the vaccines, and you know it’s really been an issue.” (NJ, Clinic Leader)

Payers also expressed concern for the negative effect COVID-19 vaccine hesitancy could have on HPV vaccine hesitancy in the future. 

“Honestly, I am really fearful that the politicization of immunization practices in general—well, the politicization of the COVID vaccine has and will continue to have some serious downstream effects on immunization uptake in childhood… I think that’s a tragedy.” (LA, Payer)

### 3.2. Inner Setting

In the CFIR inner setting domain, inner setting participants (clinic members) spoke about (1) the high motivation to counteract the low HPV vaccine rates; (2) the need for additional patient outreach and resources to facilitate HPV vaccine prioritization, and (3) the use of new vaccination strategies, such as outdoor vaccine clinics, to increase uptake during the pandemic. 

**Theme 3: Missed HPV vaccine doses due to the COVID-19 pandemic has sparked clinic motivation among staff and providers to improve uptake.** Providers spoke about their renewed motivation for HPV vaccination efforts. One provider cited that new financial incentives provided by their clinic were an additional boost to make sure eligible patients are offered the HPV vaccine. 

“I think there’s a lot of support [to increase HPV vaccine uptake] amongst my pediatric colleagues. I think people are highly motivated to get this rate up… I hear that the new quality incentive pool, this pay for performance program, that’s going to be launching as we, hopefully, emerge from COVID, is going to have a lot of adolescent measures, which is awesome because adolescent health is just such an area of need.” (LA, Provider)

**Theme 4: Limited access to wellness visits resulted in increased clinic outreach efforts to improve HPV vaccine access during the pandemic.** The COVID-19 pandemic left clinics short staffed and with lowered appointment availability. Participants described parents avoiding clinics due to a perceived risk of contracting COVID-19. 

“Well, right now with COVID, we’re impacted significantly. Our patients are certainly of the population that are more at-risk [for COVID-19].” (LA, Payer)

Clinic shutdowns and limited appointment availability limited access to wellness visit appointments. Clinic barriers coupled with a lower desire from parents to enter clinic settings during the COVID-19 pandemic reduced opportunities for HPV vaccine uptake. In response to patient fears and an effort to limit exposure to COVID-19, some clinics set up a vaccination clinic outdoors. 

“We actually stood up a vaccine tent on the outside and we outreached to people and said, ‘Hey, you’re due for these vaccines. We can set you up in a tent outside. You just have to come out for a brief second. You don’t have to walk through the hospital and get exposed to COVID.’” (LA, Provider)

Participants reported that the convenience and safety of outdoor vaccine clinics boosted HPV vaccine rates. 

The COVID-19 pandemic altered appointment availability and patient turnout for appointments. As a result, clinics had to adjust their HPV vaccine outreach and awareness efforts. 

“Because of the pandemic I have been doing outreach…I do the same thing, like I would normally do prior to the pandemic. Only difference is that I provide them with additional resources, webinars, links, webinars that they can go on and check out as well as that toolkit for them to review. But because of COVID, a lot of the focus have been on that and making sure that the kids, the adolescents, the patients are getting all of the doses that they needed or the checkups, well visits…” (NJ, Policy Representative)

### 3.3. Individuals Domain

**Theme 5: Medically underserved populations face greater challenges to receiving healthcare services, especially during the COVID-19 pandemic**. Many participants in this study described their patient population as having more challenges to receiving health services being low-income and coming from ethnically and racially diverse backgrounds. A payer participant described their diverse patient population in their safety-net clinic and the challenges some patients have making their appointments to receive all recommended vaccinations in adolescence. 

“The patient population we take care of is, even in the pediatric population, is 25% uninsured, and with CHIP [California Health Insurance Program] and the insurance programs that are available now, that’s really a marker for not being a citizen…You know, 98% of who we take care of are under 200% of the federal poverty levels as well. So, it’s a poor population. Seventy percent are best served in a language other than English and that’s predominantly Spanish. So, at a population level, it’s certainly at risk for poor health outcomes. And I would say certainly, with all the immunizations, the pandemic over the last year and a half has really been very challenging to have people come in for wellness visits and get them taken care of.” (NJ, Payer)

### 3.4. Innovation Domain 

In the innovation domain, participants described (1) adapting clinic-based strategies to improve patient experience and increase parental knowledge, (2) using community partnerships developed from the pandemic to expand HPV vaccine access (e.g., mobile van), and (3) how innovative workflow as a result of the COVID-19 pandemic, such as waiting room layout, appointment scheduling, and patient outreach, can improve HPV vaccination efforts. 

**Theme 6: Clinic-based strategies for COVID-19 mitigation and vaccination improved patients’ experiences.** Participants identified modifications to clinic procedures that took effect due to COVID-19, such as those on patient safety (e.g., social distancing, masks, separate waiting rooms for symptomatic patients) and increasing appointment turnout, as opportunities to improve HPV vaccination. One participant recalled specific strategies taken as a result of COVID-19: 

“We show how you set up an appointment, how you separate people in your waiting room, how you talk to adolescents, their parents, children, about the importance of maintenance of these [COVID-19] immunizations.” (LA, Policy Representative)

Strategies from the pandemic may support future HPV vaccine uptake efforts among adolescents. As COVID-19 rates fluctuate, waiting room safety precautions are still needed for families to feel safe and keep child wellness visit appointments where they receive adolescent vaccines, including the HPV vaccine. Vaccine communication practices used for COVID-19 vaccines around safety, testing, and importance of series completion can be applicable when describing the importance of the HPV vaccine to hesitant parents. 

**Theme 7: Clinic–community partnerships that emerged because of the COVID-19 pandemic can facilitate opportunities to increase access to HPV vaccines**. To address low COVID-19 vaccination rates among adolescents, participants reported schools reaching out to them to make the vaccine more accessible to their school-based population: 

“In March the van got parked because of COVID. We couldn’t go onsite to any of the schools. And we decided to bring everybody back to work in July, ‘cause things were calming down. We put into practice all the CDC standards to be safe on the van. And we had the schools calling us. Middle schools were calling us.” (LA, Clinic leader)

School-based partnerships may address accessibility issues and vaccination uptake with the HPV vaccine as they have with the COVID-19 vaccine [23] and may continue to improve adolescent HPV vaccination rates. 

Multilevel strategies are also needed to address HPV vaccine delay and refusal [4]. Participants in our study identified multilevel strategies (e.g., community, healthcare settings) that were used during the COVID-19 pandemic. These multilevel strategies may be beneficial in increasing HPV vaccine uptake among adolescents [16].

### 3.5. Implementation Process

The implementation process details how the innovations are implemented. Participants described (1) why and how improved clinic workflow emerged, and (2) multilevel partnerships that are needed to drive HPV vaccination success. 

**Theme 8: Identifying clinic workflow opportunities as a result of the COVID-19 pandemic.** In terms of adopting new strategies for HPV vaccination in the clinic setting, one clinic staff person mentioned: 

“Given that we’re going through COVID, and everything that’s been going on with staff shortages, I think we need to straighten our care teams here, and most importantly doing the scrubbing to really going to have the time and the staff to assist with going into these charts and identifying these patients that are eligible to get this vaccine, or any vaccine. For the most part right now, it’s getting done, but I think there’s always room for improvement to make it stronger.” (LA, Clinic Staff)

The COVID-19 pandemic affected clinic staffing shortages and staff turnover. As a result, clinic-based participants identified the need to improve workflow strategies that could improve patient care delivery, including HPV vaccine uptake. 

**Theme 9: Multilevel HPV partnerships are needed to facilitate new HPV vaccine strategies with common goals.** The pandemic strengthened many vaccination partnerships across clinics and community partners. When asked what factors would facilitate new HPV vaccine strategies, participants talked about existing partnerships to set goals and responsibilities for implementing HPV vaccine strategies. One policy participant stated: 

“I would love to see kind of like the roundtable or NJ just take on kind of …HPV vaccination goal for the state. Like, ‘As a state, we are striving for….’ … So, it’s like you’re convening the troops…So, as an advocacy group like us, ‘This is what we’re committed to doing’. This is what the hospitals are committed to doing. This is what the pediatric centers or the federal-qualified health centers are committed to doing. So, everyone knows their role, and to [having] a real strategic plan, and having …a statewide goal around HPV, and really kind of building that momentum around making it a priority…” (NJ, Policy Representative)

## 4. Discussion

While this study observed multiple factors that impeded HPV vaccine delivery during the pandemic, several innovations and partnerships in vaccine delivery were also identified as a result of the pandemic. For example, inner setting study participants navigated barriers with new outer setting community partnerships, including new bi-directional relationships with schools to support community outreach efforts and address vaccine access. By mobilizing outer setting partnerships, such as with schools and local policy and public health leaders, and adapting inner setting characteristics (e.g., staff trainings, workflow), clinics were able to build the infrastructure to increase access to COVID-19 vaccines, answer vaccine questions parents have, and train both providers and staff to better communicate with parents about the HPV vaccine. A previous study exploring challenges to HPV vaccination among adolescents found HPV vaccine efforts and quality improvement strategies in the clinic setting had fizzled in response to the pandemic [24], but inner setting participants in this study identified renewed motivation and financial incentives to address low HPV vaccine numbers.

Consistent with previous work finding politicization of the COVID-19 vaccine [25], participants in our study also identified increased politicization of COVID-19 mitigation and vaccination strategies, increasing vaccine hesitancy overall. Politicization of vaccines due to COVID-19 is a growing barrier to HPV vaccine administration in the clinic setting. Understanding factors that influence receptivity to scientific vaccine information are needed [25]. Inner and outer setting participants in our study expressed concern about the politicization of HPV vaccination and a similar pushback given that the COVID-19 vaccine was met with strong hesitancy perpetuated by misinformation on social media. To alleviate vaccine skepticism resulting from the pandemic, how channels of communication (e.g., provider and staff communication, social media, email, text messaging) can deliver trusted messages that address hesitancy resulting from the politicization of vaccines needs to be explored across communities to tailor trusted sources. 

Participants also expressed concern for reaching their patient populations with healthcare accessibility issues. Past studies have noted decreased opportunities for HPV vaccination and routine care as a barrier to HPV vaccination during the pandemic [24]. Racial and ethnic minority adolescents experience numerous barriers to receiving healthcare services [26,27,28]. The added stress of COVID-19 exposure during a clinic visit, for a population highly susceptible to more adverse outcomes [29], further increases barriers. Although past work has cited structural clinic barriers, such as limited staffing or reduced hours, that may impede HPV vaccination [24], participants in this study spoke about clinic strategies that made patients feel safer, reduced potential exposures to COVID-19, and increased convenience as effective tools for increasing HPV vaccination during the pandemic. Outdoor vaccine clinics, as an example of what was implemented during the pandemic, could be used as a future way to provide increased access to parents for their adolescents to receive the HPV vaccine. 

### Strengths and Limitations

Study strengths include interviewing participants from multiple sectors related to HPV vaccination, including policy and advocacy, to understand the complex, multilevel ways in which the pandemic impacted HPV vaccination. However, some limitations should be noted. Data were collected from one-time interviews and thus cannot assess how participant perceptions may have changed over time. This study did not investigate adolescent perspectives or hesitancy towards receiving the HPV vaccine during the pandemic. This study focused on safety-net settings serving primarily racially and ethnically diverse populations. Findings may not be generalizable to healthcare clinics serving patient populations with different sociodemographic characteristics. 

## 5. Conclusions

This study explored the clinic and community member perspectives on the impact of pandemic on HPV vaccination within safety-net primary care settings and identified emerging innovations in vaccine delivery that resulted from the pandemic. Few previous studies have thoroughly examined perspectives from inner and outer setting members on how the pandemic impacted access, delivery, and uptake of the HPV vaccine among adolescents within safety-net settings [7]. Future studies are needed to assess whether strategies influenced by the pandemic, such as new partnerships (e.g., community-based partnerships, school-based partnerships), will result in long-term implementation and sustainment for increasing HPV vaccine uptake.

## Figures and Tables

**Table 1 healthcare-11-02380-t001:** Themes organized by CFIR domains and identified as barriers and/or facilitators to HPV vaccination during the COVID-19 pandemic.

Domains	Themes
I. Outer Setting Domain (e.g., external barriers to HPV vaccination caused by COVID-19 pandemic and mitigation strategies)	Theme 1: Policy and advocacy groups collaborating with the medical community throughout the COVID-19 pandemic to improve HPV vaccination rates. (Facilitator) Theme 2: The COVID-19 vaccine has been politicized and perpetuated misinformation on social media, which may also impact HPV vaccination rates. (Barrier)
II. Inner Setting Domain (e.g., clinic barriers to HPV vaccination caused by COVID-19 pandemic and mitigation strategies)	Theme 3: Missed HPV vaccine doses due to the COVID-19 pandemic has sparked clinic motivation to improve uptake. (Barrier and Facilitator) Theme 4: Limited access to wellness visits resulted in increased clinic outreach efforts to improve HPV vaccine access during the pandemic. (Barrier and Facilitator)
III. Individuals Domain (e.g., patient characteristics, behaviors, and perceptions towards HPV vaccination that emerged from the COVID-19 pandemic)	Theme 5: Medically underserved populations face greater challenges to receiving healthcare services, especially during the COVID-19 pandemic. (Barrier)
IV. Innovation Domain (e.g., innovative strategies for HPV vaccination that emerged from the COVID-19 pandemic to improve rates)	Theme 6: Clinic-based strategies for COVID-19 mitigation and vaccination improved patients’ experiences. (Facilitator) Theme 7: Clinic–community partnerships that emerged because of the COVID-19 pandemic can facilitate opportunities to increase access to HPV vaccines. (Facilitator)
V. Implementation Process Domain (e.g., activities and strategies used for implementation of HPV vaccination as a result of COVID-19 pandemic)	Theme 8: Identification of clinic workflow opportunities as a result of the COVID-19 pandemic. (Facilitator) Theme 9: Multilevel HPV partnerships are needed to facilitate new HPV vaccine strategies with common goals. (Facilitator)

## Data Availability

Data available upon request.

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
