# Peer review of "Disruptions to and Innovations in HPV Vaccination Strategies within Safety-Net Healthcare Settings Resulting from the COVID-19 Pandemic"

_healthcare, 2023, doi:10.3390/healthcare11172380_

Round 1

Reviewer 1 Report

The level of plagiarism is 10% its good.

I have some comments:

·        Explain the procedures for selecting participants and the justification for their selection. 

·        Explain with more details about the instrument in the methodology section. 

·        Explain the validity and reliability in the methodology section. 

·        The authors must be supporting the results by previous studies.

none

Author Response

Reviewer 1

Comment 1: Explain the procedures for selecting participants and the justification for their selection. 

Response 1: We thank Reviewer 1 for the opportunity to elaborate on the methods of our study. The authors have now added additional information on the use of sampling technique and rationale for sample selection. We now reference a recent publication that details our sampling method in greater detail, Tsui et al., 2023. To avoid plagiarism, this article directs readers to a publication that providers greater detail into recruitment methods.

Tsui, J., Shin, M., Sloan, K., Martinez, B., Palinkas, L. A., Baezconde-Garbanati, L., ... & Crabtree, B. F. (2023). Understanding Clinic and Community Member Experiences with Implementation of Evidence-Based Strategies for HPV Vaccination in Safety-Net Primary Care Settings. Prevention Science, 1-16.

Comment 2: Explain with more details about the instrument in the methodology section. 

Response 2: The methodology section (lines 90-100) now include additional details on interview guide objectives guided by the Practice Change Model.

Comment 3: Explain the validity and reliability in the methodology section. 

Response 3: We thank Reviewer 1 for this suggestion. To avoid plagiarism, we now direct readers to review a previously published article (line 118-119) that thoroughly describes the coding process and how coders addressed discrepancies to ensure validity and reliability.

Comment 4: The authors must be supporting the results by previous studies.

Response 4: The authors have made substantial changes to the discussion section to incorporate how this study contributes to the growing body of literature and previous findings.

Reviewer 2 Report

Dr Samantha and colleagues interviewed people from clinic staff and from the community to investigate the impact of the COVID-19 pandemic on HPV vaccination and identified barriers and facilitators of HPV vaccination during the COVID-19 pandemic.

On the whole, a clear manuscript that can be accepted for publication. However, I have suggested the below gentle suggestions:

Gentle Comments:

Few abbreviated terms you should write in full the first time you use them, then proceed with the abbreviated form.

- Line 17; ‘HPV’ ‘Human papillomavirus (HPV)’ 

- 218; 'CHIP' 'Children's Health Insurance Program (CHIP)' 

Author Response

Reviewer 2

Few abbreviated terms you should write in full the first time you use them, then proceed with the abbreviated form.

Comment 1: Line 17; ‘HPV’ ‘Human papillomavirus (HPV)’ 

Response 1: Line 17 of the abstract now spells out the acronym before using HPV.

Comment 2: 218; 'CHIP' 'Children's Health Insurance Program (CHIP)' 

Response 2: Line 230 now includes the meaning of acronym CHIP.

Reviewer 3 Report

This is a very interesting paper and really I have not observations. I think the paper depicts the post - pandemic scenario of HPV vaccination and suggests  new vaccine approaches and partnerships that may translate to new implementation strategies for HPV vaccination within local contexts and communities. It represents a promising prospective. 

Only a formal correction: line 107, please clarify the acronym CFIR, as it appears for the first time in the text. 

Author Response

Reviewer 3

Comment 1: Only a formal correction: line 107, please clarify the acronym CFIR, as it appears for the first time in the text. 

Response 1: The meaning of acronym CFIR is now included on line 116.

Reviewer 4 Report

The manuscript submitted for review addresses an important topic. The research idea is very interesting, and the results presented, and their structure is very interesting. However, the manuscript needs to be refined in terms of methodology, presentation of results, and discussion. 

It is a research with a great potential for good paper. 

Keywords

L. 34-35 To increase the scope of the paper, I suggest not repeating Keywords words that are already in the title of the manuscript. Instead, I suggest using words that relate to the methodology of the paper, e.g., Qualitative research, Focus group

Introduction

L. 48-50 It is helpful to specify that the data refer to the U.S.

L. 51-52 I think it is helpful to refer to the literature and perhaps briefly describe how the authors understand the terms "safety-net healthcare settings" - as there may be a different understanding

https://pubmed.ncbi.nlm.nih.gov/24164728/

L 54-55 Isn't that what the sentences above L 41-43 refer to? Its theme recurs in the sentence L 64-66 in different wording. Please modify the structure of the Introduction. Its current layout gives the impression of several loose thoughts and loops the thread of the impact of COVID-19 on HPV vaccination implementation. 

L 70-73 Did the study's authors pose any research questions they found answers to while implementing the study?

Materials and Methods

L 76 If the authors want to be methodologically precise, they did not use mixed methods but mixed techniques as part of the qualitative method in sociomedical research.

L 77 What does the term "multilevel participants" mean?

L 80 and L 19 (Abstract) If the study's authors conducted focus group interviews within seven groups, then n is not equal to 7 because the number n refers to the participants involved in the study, not the number of studies conducted. Thus, the number n for Focus groups is the number of all participants who participated in the FGIs.

L 86-89 The information in these sentences should form the Ethical Considerations subsection, which should also include information on how the confidentiality of the research was guaranteed to respondents and how access to the meeting recordings was protected. 

What I find missing in this section of the manuscript is information about the tools, or interview questionnaires, used in the study. There is only the information "semi-structured interview" L 93. What issues were paid attention to when conducting IDIs, and what issues were paid attention to when conducting FGIs? For what reason did the authors of the study decide to use these two techniques - for what reason? Which research questions were supported by the IDI technique and which by the FGI technique? 

Results

The Results section is not very clear. It may be due to the fact that Table 1 - which explains a lot - is only at the end of the chapter. 

Since there is no information in the manuscript about the research questions and issues in the interview questionnaires, it is not entirely clear whether the L 110 "nine themes emerged" during the interviews or whether it was the nine themes that were asked about during the interviews. 

The Results chapter makes interesting use of the data collected from the IDIs, although the very modest number of citations from the 58 interviews leaves much to be desired. 

I need help finding references to focus group data on this chapter. If parents of children appear, it is mainly in the stories of specialists. It once again raises the question: what was the purpose of conducting the FGIs?

Discussion

The first part of the Discussion is very interesting, although the first part summarizes the Results. However, in the second part, when the authors discuss the L 325-326 patients' barriers, I need to figure out what to discuss - although the topic is very interesting - because there is no discussion of the Results in the Results. 

Author Response

Reviewer 4

Keywords

Comment 1: L. 34-35 To increase the scope of the paper, I suggest not repeating Keywords words that are already in the title of the manuscript. Instead, I suggest using words that relate to the methodology of the paper, e.g., Qualitative research, Focus group

Response 3: We thank reviewer 3 for this suggestion. Keywords ‘HPV vaccination’ and ‘COVID-19 pandemic’ have been replaced with ‘Qualitative Research’ and ‘depth interviews’.

Introduction

Comment 2: L. 48-50 It is helpful to specify that the data refer to the U.S.

Response 2: Line 47-50 now refers to data from the U.S. This line now reads, “Between March and August 2020, US HPV vaccine initiation rates were only 23% of rates in 2018 and 2019 during the same time frame [5].”

Comment 3: L. 51-52 I think it is helpful to refer to the literature and perhaps briefly describe how the authors understand the terms "safety-net healthcare settings" - as there may be a different understanding

https://pubmed.ncbi.nlm.nih.gov/24164728/

Response 3:  We thank Reviewer 4 for this comment. We now define “safety-net healthcare settings” on lines 53-55.

Comment 4: L 54-55 Isn't that what the sentences above L 41-43 refer to? Its theme recurs in the sentence L 64-66 in different wording. Please modify the structure of the Introduction. Its current layout gives the impression of several loose thoughts and loops the thread of the impact of COVID-19 on HPV vaccination implementation. 

Response 4: We thank Reviewer 4 for this comment. The authors have now modified the introduction to more clearly structure our background description of the impact of COVID-19 on HPV vaccination implementation by describing structural barriers, individual barriers and multilevel evidence-based strategies to addressing barriers in the clinic setting.

Comment 5: L 70-73 Did the study's authors pose any research questions they found answers to while implementing the study?

Response 5: We thank Reviewer 4 for this suggestion. Lines 74-76 now state “Our research question was: How did the COVID-19 pandemic impede and/or facilitate vaccination strategies that can support HPV vaccination among adolescents?”

Materials and Methods

Comment 6: L 76 If the authors want to be methodologically precise, they did not use mixed methods but mixed techniques as part of the qualitative method in sociomedical research.

Response 6: Lines 79-82 now read, “This study used qualitative study is part of a larger mixed methods study funded by the National Cancer Institute (R37CA242541), where we recruit clinic team members and parent participants to investigate multilevel experiences with EBS for HPV vaccination within safety-net primary care settings to inform quantitative methods to implement EBS.”

Comment 7: L 77 What does the term "multilevel participants" mean?

Response 7: This line has now been rewritten to read, “This study used qualitative study is part of a larger mixed methods study funded by the National Cancer Institute (R37CA242541), where we recruited clinic and community level participants to investigate experiences with EBS for HPV vaccination within safety-net primary care settings.”

Comment 8: L 80 and L 19 (Abstract) If the study's authors conducted focus group interviews within seven groups, then n is not equal to 7 because the number n refers to the participants involved in the study, not the number of studies conducted. Thus, the number n for Focus groups is the number of all participants who participated in the FGIs.

Response 8: We thank Reviewer 4 for this comment. In addressing Comments 10 and 13, the author’s have decided to remove focus group data as part of the analysis.

Comment 9: L 86-89 The information in these sentences should form the Ethical Considerations subsection, which should also include information on how the confidentiality of the research was guaranteed to respondents and how access to the meeting recordings was protected. 

Response 9: The authors thank Reviewer 4 for this suggestion. We have now added an ethical considerations subsection (2.3) starting on line 103. Information on protection of participant confidentiality have now been included.

Comment 10: What I find missing in this section of the manuscript is information about the tools, or interview questionnaires, used in the study. There is only the information "semi-structured interview" L 93. What issues were paid attention to when conducting IDIs, and what issues were paid attention to when conducting FGIs? For what reason did the authors of the study decide to use these two techniques - for what reason? Which research questions were supported by the IDI technique and which by the FGI technique? 

Response 11: We thank Review 4 for attention to this detail. The authors have removed focus group data from this paper and now only focus on IDIs.

Results

Comment 11: The Results section is not very clear. It may be due to the fact that Table 1 - which explains a lot - is only at the end of the chapter. 

Response 11: We thank Reviewer 4 providing this feedback. Tables appearing at the end of the chapter are aligned with template provided by the journal. The authors believe addressing Comment 5 and Comment 10, regarding our research question and objectives of IDIs improved readability of the results section.

Comment 12: Since there is no information in the manuscript about the research questions and issues in the interview questionnaires, it is not entirely clear whether the L 110 "nine themes emerged" during the interviews or whether it was the nine themes that were asked about during the interviews. 

Response #12: In addressing comment #5, we have now included our research question to the manuscript (lines 74-76) and believe this will bring clarity to our results sections. Section 2.2 on data collection provides an example open-ended question that provides clarity on how 9 themes emerged.

Comment 13: The Results chapter makes interesting use of the data collected from the IDIs, although the very modest number of citations from the 58 interviews leaves much to be desired. 

I need help finding references to focus group data on this chapter. If parents of children appear, it is mainly in the stories of specialists. It once again raises the question: what was the purpose of conducting the FGIs?

Response 13: We thank Reviewer 4 for this comment. We conducted focus groups to assess if there was variation in perspectives between vaccinated and unvaccinated parents among racial and ethnic minority parents. We did not observe differences in HPV vaccine barriers due to COVID-19. Since the focus of this paper is to understand clinic disruptions due to the COVID-19 pandemic and parent focus groups were not centered on the clinic barriers, the authors have chosen to remove focus group data from this analysis in this study.

Discussion

Comment 14: The first part of the Discussion is very interesting, although the first part summarizes the Results. However, in the second part, when the authors discuss the L 325-326 patients' barriers, I need to figure out what to discuss - although the topic is very interesting - because there is no discussion of the Results in the Results. 

Response 14: We thank Reviewer 4 for the opportunity to improve the discussion section, paragraph 2. We had limited discussion of patient level barriers because the scope of our work was focused on clinic level barriers. However, we acknowledge the importance of patient level barriers among the medically underserved populations that were exacerbated during the pandemic by citing relevant works that may overlap with clinic level barriers. This section has been revised to better focus on clinic barriers to delivering the HPV vaccine post-pandemic.

Round 2

Reviewer 4 Report

Thank you.

Author Response

Thank you for reviewing our manuscript.